# Implementing Toll Road Infrastructure Financing in Indonesia: Critical Success Factors from the Perspective of Toll Road Companies

**Muhammad Fauzan [1], Heri Kuswanto [2,]\* and Christiono Utomo [3]**

1  Interdisciplinary School of Management Technology, Institut Teknologi Sepuluh Nopember,
   Jl. Arif Rahman Hakim, Surabaya 60111, Indonesia; fauzanmm@gmail.com
2  Statistics Department, Faculty of Science and Data Analytics, Institut Teknologi Sepuluh Nopember, Jl. Arif
   Rahman Hakim, Surabaya 60111, Indonesia
3  Civil Engineering Department, Institut Teknologi Sepuluh Nopember, Jl. Arif Rahman Hakim, Surabaya
   60111, Indonesia; christiono@ce.its.ac.id
\*  Correspondence: heri_k@statistika.its.ac.id

**Abstract:** Having effective and efficient financing is one of the most critical steps in accelerating public infrastructure development, including toll roads. This study aims to identify critical success factors (CSFs) for implementing toll infrastructure financing in Indonesia. Thirty-three CSFs have been identified from the literature review. A Delphi survey was conducted involving a panel of experts in the infrastructure industry. Based on the survey, it is known that the internal rate of return, affordability, investment decisions, commercial banks, financing costs, interest rate risk, control of cash flow, contract scope, and principles of risk transfer are important factors for implementing toll infrastructure financing in Indonesia. This study fills research gaps by developing a CSF model for successful toll road infrastructure financing in Indonesian PPPs, considering private perspectives and aiming to provide insights for investors and enhance understanding of country profiles in developing countries. The focus on toll road implementation in Indonesia contributes to a comprehensive understanding of CSFs for PPPs in the country.

**Keywords:** critical success factors; financing implementation; Indonesia toll roads; business entity perspective

## 1. Introduction

The economic progression of developing regions in Asia can be attributed to improvements in infrastructure. Nevertheless, insufficient investment poses obstacles, including a marked insufficiency in access to physical infrastructure and services, particularly in economically disadvantaged regions. Across Asia, countless individuals grapple with significant difficulties in accessing vital services. In 2019, around 770 million lacked electricity access, primarily in sub-Saharan Africa and South Asia (World Bank 2021). By 2021, about 785 million still lacked basic drinking water services, with more lacking safe water sources, while 2.2 billion lacked safe sanitation facilities (WHO 2019). Even among those with some degree of access, the quality of services could be better, affecting both rural and urban areas (Bouraima et al. 2020). A wide range of problems contribute to these poor conditions. Issues like irregular power supply interfere with productivity and disrupt people's day-to-day lives. Overcrowded roads and ports obstruct efficient trade and transport, constraining economic advancement and opportunities (Nur et al. 2023). Moreover, inferior water and sewage systems lead to unsanitary environments, presenting health threats to local communities. Poor-quality schools and healthcare establishments further amplify these challenges, depriving individuals of vital education and medical care (Swarnakar et al. 2022). These deficiencies are mirrored in the global rankings of

many developing Asian economies. As per Schwab and Sala-i-Martin (2017), these nations trail behind in infrastructure development, underlining the pressing need for investment and enhancement in this vital sector. Infrastructure, fundamental to a nation's economic trajectory, bridges market efficiencies, industry support, and trade facilitation. In developed countries, advanced infrastructure paves the way for diversified economies, elevated GDPs, and sustainable urbanization. Conversely, many developing nations grappling with inadequate infrastructure often witness constrained growth, reduced foreign investments, and urbanization woes. Infrastructure is much more than bricks and mortar. It is the foundation upon which economies grow, societies thrive, and nations compete globally (Asian Development Bank 2012).

Filling these infrastructure gaps is crucial for improving living conditions and fostering sustainable regional growth. The shortage of infrastructure in developing Asia stems from restricted financial resources and effective methods for resource allocation. Despite the acknowledged need for infrastructure advancement, stringent fiscal conditions and limited public sector capability impede progress in bridging this gap. One proposed solution involves enlisting the private sector for infrastructure development, utilizing their expertise in operational efficiency, financing, innovation, and skills (Laishram and Kalidindi 2009). Reimagining the collaboration between the private and public sectors through public–private partnerships (PPPs) can improve the adequate provision of public goods and services. PPPs represent long-term contracts where private entities and government bodies collaborate, with the private sector taking on substantial risk and managerial responsibility in exchange for performance-based compensation. The investment in PPPs has the potential to address these ongoing infrastructure challenges.

Public–private partnerships (PPPs) offer a novel approach to infrastructure development, particularly in emerging and developing nations, like for road projects. Governments use PPPs to leverage the private sector's expertise, innovation, and management, ensuring project efficiency (Abdel Aziz 2007). Additionally, governments, especially in developing countries, use PPPs to supplement their budgets with private financing, given the other fiscal demands (Kumaraswamy and Zhang 2001). PPPs represent a blend of public accountability and private sector efficiency. They enable better risk management, faster project execution, and harness private innovation, often resulting in enhanced maintenance and operational efficiency. The joint responsibility in PPPs ensures quality and sustainable outcomes. However, PPPs have their downsides. Companies might aim for profit without genuine effort, possibly through favorable government contracts. They might sway officials to frame contracts that do not serve public interests. Numerous stakeholders, from corporations to environmentalists, push for PPP projects, occasionally placing particular interests above those of the public (Borman and Janssen 2013). Bureaucrats, motivated by personal interests, significantly influence PPP outcomes. Politicians might favor projects for electoral benefits rather than a broader societal good. The private sector's deeper insights can lead to imbalanced risk and reward sharing. Moreover, the political landscape, encompassing governance and transparency, profoundly impacts the success and fairness of PPPs (Gawel 2017).

Since the onset of the 21st century, financing public–private partnerships (PPPs) in key Southeast Asian nations, such as Indonesia, Malaysia, the Philippines, Thailand, and Vietnam, has remained at under 1% of their yearly GDP (Zen 2018). The deterrent of canceled projects, which result in large unrecoverable expenses, cannot be ignored. Between 1991 and 2015, abandoned PPP ventures represented USD 41.6 billion in initial pledged investment, impacting 6.3% of all dedicated PPP investments in developing Asia (Deep et al. 2019). The efficacy of enduring infrastructure PPP schemes hinges on the financing frameworks established. The utilization of project finance, a strategy that involves forming a discrete legal and economic body to oversee the project and procure necessary fiscal resources, is pivotal to the functioning of PPPs. Considering the risks that are inherent in sizeable PPP deals, project finance ensures that risks and their associated returns are aligned with the entities who are most competent at handling them. This encourages investor cooperation by enabling a fair and rational allocation of risk. Additionally, project

finance allows for extended-term debt, which is crucial for covering substantial capital costs. Moreover, leveraging project finance can help tackle issues of information disparity that are frequently faced in large infrastructure PPP projects. By alleviating these concerns, project finance aids in the seamless execution of such initiatives. In advanced economies, a wealth of financiers and strong support networks exist for public–private partnerships (PPPs). However, in emerging Southeast Asian nations, the infrastructure industry ecosystem struggles due to insufficient stakeholders. For instance, the domestic currency bond markets in nations such as Indonesia, Malaysia, the Philippines, Singapore, Thailand, and Vietnam need to be more recognized than their counterparts in Japan and other developed economies (Deep et al. 2019). This constrained financial capability impedes the smooth operation of the infrastructure sector in these regions. The effective execution of infrastructure PPP initiatives can differ significantly across various sectors.

Conversely, toll road initiatives, particularly those with environmental sustainability aspects, necessitate comprehensive evaluation procedures and precise demand forecasts. When a project is urgently required, the most immediate strategy often involves public procurement or delegating a state-owned enterprise (SOE) to execute the project. In Indonesia, numerous infrastructure ventures are undertaken by SOEs via direct assignment. With the world's fourth largest population, following China, India, and the United States (Kurniawati and Sugiyanto 2021), Indonesia is expected to experience a demographic bonus, with a significant proportion of its population entering the productive age bracket (Wibowo and Alfen 2015). Nevertheless, the quality of human resources and infrastructure provision could be better, constituting significant challenges. Despite these obstacles, the government has launched programs in education, health, and infrastructure, resulting in a substantial increase in the assigned budget from IDR 117 trillion in 2014 to IDR 417 trillion in 2020. This is a positive indicator for Indonesia's economic future, with PwC (2017) forecasting that Indonesia will become the fifth-largest global economy by 2030 and the fourth-largest by 2050.

Indonesia is facing a concerning budget deficit due to the increased expenditure on infrastructure, surpassing tax revenue growth. Additionally, the budget allocation is primarily based on input parameters rather than performance metrics, hindering its effectiveness in achieving developmental goals (Wibowo and Alfen 2015). To address these challenges, the government should explore alternative financing models beyond the traditional Capital States scheme, such as public–private partnerships (PPPs). PPPs offer several advantages, including improved spending quality, reduced strain on the Capital States, a direct link between budgeting and performance metrics, and the promotion of innovative approaches to enhance public service quality (Sharma et al. 2010). A PPP is a collaborative effort between the government and private sector entities involving long-term contracts and private financing for design, construction, and operation. The government or users provide payments throughout the contract term, and ownership transfers from the private sector to the government upon contract completion (Chen 2020). Adopting PPPs is an effective strategy for creating value in public infrastructure projects, particularly in large-scale construction initiatives that require substantial investments (Chen et al. 2015). In Indonesia, PPP initiatives have been implemented since the New Order era, primarily in toll roads and the electricity sectors. However, the significant development of PPPs began after the financial crisis in 1998. Presidential Regulation No. 38/2015 introduced the PPP concept, defining it as a partnership between the government and business entities to provide public-interest infrastructure, leveraging resources from these entities, with shared risk among all involved parties. The build–own–operate–transfer (BOOT) contract is the most commonly used form in PPPs, offering an optimal risk-sharing mechanism between the government and business entities (Chen et al. 2015). The choice of the PPP model depends on risk analysis, financial analysis, and legal considerations, with risks identified, allocated, and managed by the parties incurring the lowest cost. Critical risks in PPPs include land acquisition, profit repatriation, infrastructure construction and operation, commercial viability, and legal certainty (Hoppe and Schmitz 2013).

The government can enhance the attractiveness of PPP projects to investors by providing financial contributions, including grants, loans, subsidies, and sharing risks with investors. Additionally, offering a minimum revenue guarantee for the BOT project can alleviate market risks and stimulate investor engagement (Wibowo and Alfen 2015). An ideal capital structure for PPP ventures is a blend of debt and equity that augments the firm's value (Brigham and Ehrhardt 2016). Before defining the ideal capital structure, the government predetermines the tariff subsidy and other support in the concession agreement. The consideration of social capital is required in calculating economic feasibility and holistic project financing, reflecting the project's cash flow risks. Efficient cash flow management is fundamental for the viability and steadiness of the sustainable construction sector, as liquidity is a crucial resource for a successfully operating project organization (Ke et al. 2017). The collaboration mechanism for infrastructure projects in Indonesia allows government or business entities to make proposals. Government-proposed projects are included in the priority list, publicly disclosed, and accessible to everyone. Business entities can also propose projects for collaboration with the government, provided that specific criteria are met. The government assesses the project's feasibility and proceeds with a public tender, granting compensation for accepted projects.

In Indonesia, state-owned enterprises (SOEs) are tasked as development agents to ensure economic parity. SOEs are business entities where the state owns the capital, crucial in managing the national economy toward societal welfare. However, the execution of SOEs' role has not been ideal. This concern is both philosophical and sociological regarding the existence of SOEs. This is because, in some instances, SOEs have been more efficient and effective in their operations than the government. Furthermore, SOEs can generate revenue for the government through their business activities. Nevertheless, it is crucial to remember that Indonesian SOEs should not exclusively prioritize profit maximization but also acknowledge their role in delivering public services and contributing to societal well-being. Being state-owned entities, they must serve the public interest and facilitate the nation's development. In conclusion, SOEs play a pivotal role in fostering societal welfare through their engagement in various economic sectors and commitment to providing public services. Although they also have a role in generating government revenue, their public service responsibility should not be overshadowed. This investigation was focused on a major infrastructure project in Indonesia, specifically the BOT toll project across Java and Sumatra island, identified as part of the National Strategic Project.

This research aims to fill the gap regarding identifying the critical success factors that are required for financing toll road infrastructure. Specifically, it aims to craft a comprehensive CSF model from the vantage point of local toll road companies, providing invaluable insights for investors in PPP-based infrastructure projects and enhancing the comprehension of Indonesia's unique context. While previous studies have touched upon CSFs in general PPP projects, a discernible gap exists in understanding the nuances that are specific to Indonesian toll roads. This research delves deep into this niche, highlighting that toll roads bolster the nation's economic tapestry beyond merely serving as transportation routes, catalyzing economic growth, trade, urban efficiency, and offering a sustainable revenue source. Through this, it emphasizes the imperative of ensuring their successful implementation for broader infrastructural advancement.

## 2. Literature Review

### 2.1. Public–Private Partnerships (PPP)

Bing et al. (2005) define PPPs as a long-term contractual arrangement between a public sector agency and a private sector concern, whereby resources and risk are shared to develop a public facility. The principal aim of a PPP for the public sector is to achieve value for money in the services provided while ensuring that the private sector entities meet their contractual obligations properly and efficiently (Grimsey and Lewis 2004). PPPs are a means of public sector procurement using private sector finance and best practice. PPPs can involve designing, constructing, financing, operating, and maintaining public infrastructure

and facilities or services to meet public needs. They are often privately financed and operated based on revenues received for the delivery of the facility and/or services. One key to this is the ability of the private sector to provide more favorable long-term financing options that may be available to a government entity and to secure the financing in a much quicker time frame (NCPP 2003). Such contracts are long-term in nature and typically last 25–30 years. PPPs address the common faults associated with public sector procurement, such as high construction costs, construction overruns, operational inefficiencies, poor design, and community dissatisfaction. The PPP is founded on transferring risk from the public to the private sector under circumstances where the private sector is best placed to manage risk. One of the critical features of the PPP, which is appealing to the government, is the shift of project risks from the public sector to the consortium involved with the project, even though this requires a profit incentive to the project consortium (Grimsey and Lewis 2004). PPPs are being established as a cost-effective method of overcoming the costs associated with the provision and maintenance of infrastructure.

### 2.2. PPPs in Indonesia

Indonesia has proactively explored public–private partnerships (PPPs) to bridge its pronounced infrastructure investment gap, the largest among the G20 nations, as per Woetzel et al. (2016). With an infrastructure demand reaching a staggering USD 369 billion between 2015 and 2019 for roads, transport, and electricity, the available funds from central and local governments covered just 41% of this need. The government aimed to fill this 59% investment shortfall primarily through PPPs, targeting contributions of USD 135 billion (Endo et al. 2021). Several initiatives, such as the Sarana Multi Infrastructure (SMI) in 2009 and the Indonesia Infrastructure Guarantee Fund (IIGF), were established to promote PPPs. However, despite the challenges, Indonesia's PPP environment is considered one of the most developed in the ASEAN region, rivaling the likes of the Philippines, Thailand, and Vietnam (Asian Development Bank 2012). As the OECD insightfully commented, the road to infrastructure development is paved with challenges, but the confluence of public ambition and private diligence can shape the path forward.

### 2.3. Critical Success Factors (CSFs)

The concept of critical success factors (CSFs) was developed by Rockart and the Sloan School of Management, with the phrase first used in the context of information systems and project management (Rockart 1982). Jefferies et al. (2002) state that critical success factors are those fundamental issues, inherent in the project, which must be maintained for team working to occur efficiently and effectively. They require day-to-day attention and operate throughout the life of the project. This section describes some CSFs related to toll road infrastructure financing. Based on the literature review, the factors influencing the success of toll road infrastructure financing can be explained in Table 1.

**Table 1.** Critical success factors of toll road infrastructure Financing.

| Code | Factors | References |
|------|---------|-----------|
| Investment Analysis | | |
| IA1 | Net Present Value (NPV) | Zala and Vel (2019); Ameyaw and Chan (2015); Ashuri et al. (2010) |
| IA2 | Discounted Cash Flow (DCF) | de Albornoz et al. (2021); Warner (2013); Vassallo et al. (2012); (Warner 2013) |
| IA3 | Internal Rate of Return (IRR) | de Albornoz et al. (2021); Vassallo et al. (2012) |
| IA4 | Payback Period | de Albornoz et al. (2021); Vassallo et al. (2012) |
| IA5 | Profitability to attract investors and lenders | Chou and Pramudawardhani (2015); Gupta et al. (2013) |
| IA6 | Good feasibility studies | Chou and Pramudawardhani (2015); Gupta et al. (2013); Dulaimi et al. (2010); Jefferies (2006); (Dulaimi et al. 2010) |

**Table 1.** *Cont.*

| Code | Factors | References |
|------|---------|-----------|
| Public sector investment decision | | |
| PID1 | Business Diversification | Almarri and Boussabaine (2017) |
| PID2 | Economic Justification | Jayasena et al. (2020); Yescombe and Farquharson (2018) |
| PID3 | Affordability | Yescombe and Farquharson (2018); Jacobson and Choi (2008); Olusola Babatunde et al. (2012) |
| The Private Sector Investor's Perspective | | |
| PIP1 | The Investment Pool | Tang et al. (2012); Wang et al. (2007) |
| PIP2 | The Investment Decision | Yescombe and Farquharson (2018); Jefferies (2006) |
| PIP3 | Joint-Venture Issues | Yescombe and Farquharson (2018); Jefferies (2006); Bing et al. (2005); Zhang (2005) |
| Private Sector Financing—Sources and Procedures | | |
| FPSP1 | Commercial Banks | Ameyaw and Chan (2016); Hwang and Lim (2013); Xu et al. (2010) |
| FPSP2 | Bond Issues | Ameyaw and Chan (2016); Hwang et al. (2013); Xu et al. (2010) |
| FPSP3 | Availability of financial markets | Liu et al. (2021); Malek and Gundaliya (2021); Lam and Yang (2020); Ke et al. (2010) |
| Financial Structuring | | |
| FS1 | The Financial Model | Yao et al. (2018); Yescombe and Farquharson (2018); Regan (2012) (Regan 2012) |
| FS2 | Model Inputs and Outputs | Yescombe and Farquharson (2018); Ameyaw and Chan (2016) |
| FS3 | Financing Costs | Yescombe and Farquharson (2018, 2013) |
| FS4 | Debt Profile | Yescombe and Farquharson (2018); Ismail (2013) |
| Financial Hedging | | |
| FH1 | Interest Rate Risk | Cheung and Chan (2011); Gholamreza and Zeinab (2012) |
| FH2 | Inflation Issues | Gholamreza and Zeinab (2012); Cheung and Chan (2011); Jie and Zou (2011); Xu et al. (2010); Boeing Singh and Kalidindi (2006) |
| FH3 | Exchange Risk | Chou et al. (2012); Cheung and Chan (2011); Ke et al. (2010) |
| Lenders' Cash-Flow Controls, Security, and Enforcement | | |
| LCSE1 | Control of Cash Flow | Yescombe and Farquharson (2018); Liu et al. (2020) |
| LCSE2 | Security | Liu et al. (2020); Yescombe and Farquharson (2018) |
| LCSE3 | Intercreditor issues | Liu et al. (2020); Yescombe and Farquharson (2018) |
| Service-Fee Mechanism | | |
| SFM1 | Contract Scope | Chou and Pramudawardhani (2015); Hwang and Lim (2013); Xu et al. (2010) |
| SFM2 | Payment and Scheme Structure | Chou et al. (2012); Cheung and Chan (2011); Ke et al. (2010); Xu et al. (2010) |
| SFM3 | Third-Party and Secondary Revenues | Yescombe and Farquharson (2018) |
| Risk Evaluation and Transfer | | |
| RET1 | Principles of Risk Transfer | Chou and Pramudawardhani (2015); Hwang and Lim (2013); Bing et al. (2005) |
| RET2 | Political Risks | Jefferies (2006); Zhang (2005) |
| RET3 | Construction Risks | Bing et al. (2005); Jin (2010); Ke et al. (2010); Xu et al. (2010); Hwang and Lim (2013) |
| RET4 | Completion Risks | Hwang and Lim (2013); Chou et al. (2012); Jin (2010); Bing et al. (2005) |
| RET5 | Operation-Phase Risks | Hwang et al. (2013); Chou et al. (2012); Gholamreza and Zeinab (2012); Cheung and Chan (2011); Ke et al. (2010); Bing et al. (2005) |

### 3. Research Method

Qualitative methods were used to answer questions about implementing toll road infrastructure financing in Indonesia. In this method, a literature review is carried out regarding the funding models that are currently used for toll roads. The process of determining the funding model is carried out to obtain toll road funding model variables and then obtain the factors that influence and define the model. After obtaining the funding model variables, the process of determining the most appropriate funding model is carried out by conducting in-depth interviews so that the concept of a toll road funding model will emerge. Apart from qualitative methods in the form of literature studies, interviews, and document collection, literature studies are also carried out to uncover similar research. This is undertaken to help provide an overview of the methods and techniques used in research.

After obtaining the funding model that best suits the interview results, the next step is to obtain the critical success factors that must be considered to implement the model for implementing toll road infrastructure financing in Indonesia. Therefore, the CSF method was carried out to answer the determination/development of the selected funding model method/scheme so that it can be applied to toll roads in Indonesia. CSFs are a limited number of areas that are important for achieving success, either in the context of an organization or project implementation (Amponsah and Forbes 2012). By identifying the selected CSF funding model, it can be understood how the selected funding model can be successfully implemented. The steps taken are identifying success factors from studies of the literature, developing a comprehensive questionnaire and testing the questionnaire, collecting data through interviews, conducting data analysis, and obtaining CSFs from the selected funding (Li et al. 2005). In the end, validation was carried out on the results of implementing toll road infrastructure financing in Indonesia. Then, conclusions and suggestions were made to develop a toll road funding implementation model using this model in Indonesia to answer the research objectives.

This complete list of success factors was identified from previous studies. After identifying the success factors of the selected funding model, a comprehensive questionnaire was developed, and the questionnaire was tested so that respondents could understand the questionnaire. The questionnaire was designed using the Delphi survey method, as seen in Figure 1. Data analysis from the Delphi method carried out in this research was divided into two rounds, namely the first round of data analysis and the second round of data analysis.

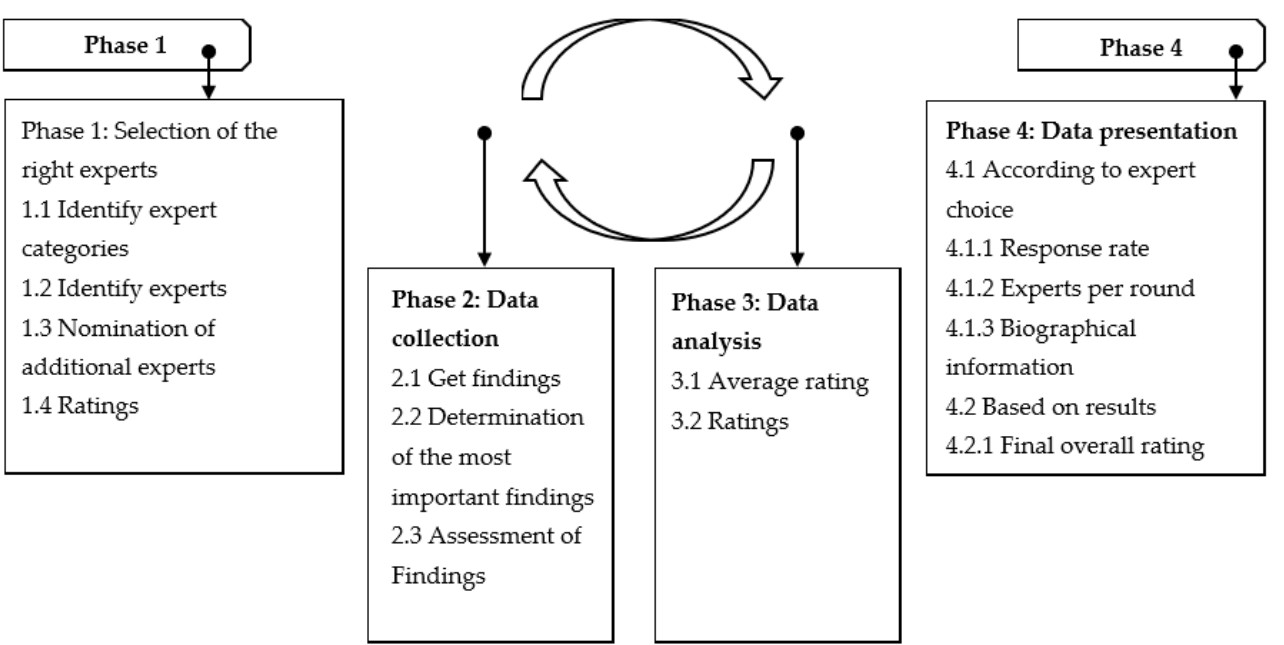

**Figure 1.** Model Delphi survey.

The Delphi method is designed to gather the most dependable expert opinions collectively. It works through an anonymous multi-step survey system, using group feedback as a check after each stage (von der Gracht 2012). This technique is frequently used in management research to gather data on intricate situations, especially when a shared or aligned expert viewpoint can offer deep insights (Hallowell and Gambatese 2009; Hsu and Sandford 2007). Given its efficacy, the Delphi method is chosen for tackling intricate issues, such as financing toll road infrastructures in Indonesia. This involves querying experts in various survey rounds. After each round, feedback from these experts is anticipated (Dalkey and Helmer 1963). The process of the Delphi method can be seen in Figure 1.

The critical success factors (CSFs) approach is a powerful method for identifying key elements that are vital for effective operation, from managing time and processes to achieving desired results (Borman and Janssen 2013). Daniel introduced the concept of success factors in the 1960s, later expanded by Rockart (1982), who classified CSFs as crucial activity areas that guarantee favorable outcomes and assist in meeting specific management objectives. Pinto and Slevin's work in 1987, known for providing a comprehensive list of success factors, has been particularly influential (Liu et al. 2015). Definitions of CSFs vary across the scholarly literature. Pinto and Slevin, for instance, concentrated on factors that significantly boost the chances of successful project execution (Pinto and Slevin 1987), while Maghsoodi and Khalilzadeh stressed the project management inputs that directly lead to project success (Maghsoodi and Khalilzadeh 2018). According to Hofer and Schendel, cited by Leidecker and Bruno (1984), CSFs are variables under managerial control that significantly shape a company's competitive standing within an industry.

On the other hand, Leidecker and Bruno view CSFs as features, conditions, or variables that, when well managed or maintained, can significantly impact a company's industry success. Amberg (2005) suggested that Rockart's approach remains particularly relevant in project management. Furthermore, the Delphi technique is commonly used for identifying factors in research. It fosters effective group dynamics via an anonymous, multi-step survey process that uses group feedback as a control mechanism after each round (von der Gracht 2012). Delphi surveys have been extensively used in management research for empirical data collection, particularly in situations requiring complex modeling where expert opinion consensus or convergence is vital (Hallowell and Gambatese 2009). The research methodology for this study included two key stages: first, initial variable identification based on an extensive literature review, and second, a survey questionnaire to gather responses about the uncertainty factors contributing to the CSFs for toll road infrastructure financing in Indonesia. The research began with a preliminary list of factors and a literature review identifying 33 factors. The final step involved designing a questionnaire and defining a data collection method. The questionnaire had two parts: one to collect demographic information and the second to gauge the respondents' level of agreement with each uncertainty factor. Participants were instructed to use a Likert scale ranging from one to six to express their views on the significance of specific factors, where "6" denotes extremely important, "5" is important, "4" is slightly important, "3" is slightly less important, "2" is less important, and "1" is negligible. The researchers profiled the respondents, determining the target population and sample size for different regions and restricting the sample to individuals from public sector organizations. The respondents were chosen based on their experience with large and small construction projects, and a simple random sampling method was used. The respondents were given a day to complete the survey to ensure thorough data collection.

As shown in Table 2, most respondents had 5 to 10 years of experience, and there was significant representation from business entities and financial institutions. Data were gathered from major toll road businesses in Indonesia, including PT Adhi Karya (Persero) Tbk, PT Hutama Karya (Persero), PT Pembangunan Perumahan (Persero) Tbk, PT Waskita Karya (Persero) Tbk, and PT Wijaya Karya (Persero) Tbk. These organizations, having large-scale operations and numerous nationwide projects, provided potential respondents, and data were collected using self-administered methods.

**Table 2.** Basic information of respondents.

| Description | Type | Frequency | Percentage (%) |
|---|---|---|---|
| Field of expertise | Engineering | 27 | 28 |
| | Financial | 56 | 59 |
| | Facility management | 9 | 9 |
| | Others | 3 | 3 |
| Position PPP project experience | Management | 43 | 45 |
| | Undertaker | 39 | 41 |
| | Others | 13 | 14 |
| Position | Less than 6 years | 12 | 13 |
| | 6–15 years | 29 | 31 |
| | 16–30 years | 31 | 33 |
| | More than 30 years | 23 | 24 |

The study employed an organizational perspective, gathering data from public sector construction industry respondents who were involved in small and large projects. Primary data were gathered through questionnaires distributed in Indonesia from October to November 2022, with 97 questionnaires sent to PPP practitioners through direct mail, email, and online platforms. Of these, 95 were returned, leading to a 98% response rate. After considering the valid questionnaires, the effective data response rate was similar. As demonstrated in Table 2 of the research article, respondents had significant experience in infrastructure projects, with 89% having over six years of professional experience.

## 4. Discussion

### 4.1. Data Analysis and Results

Data analysis from the Delphi method carried out in this research was divided into two rounds, namely the first round of data analysis and the second round of data analysis.

#### 4.1.1. First Round Data Analysis

In the first round, the number of respondents was 95 selected sources, representing the toll company sector in Indonesia. Each question was answered on a Likert scale of one to six, and the procedure is explained in Section 3. All respondents had experience in the toll road sector, and the average experience was more than ten years. In addition, around 57% of respondents had more than 15 years of experience.

The relative importance index (*RII*) was commonly used for data analysis in previous research. For instance, El-Sayegh utilized these tools to assess risk and allocate it in the construction industry of the United Arab Emirates (UAE). El-Sayegh and Mansour (2015) also applied these techniques to examine risk in UAE highway infrastructure projects. In line with these studies, the present research adopted the same approach to investigate critical success factors for toll road infrastructure financing implementation in Indonesia. To determine the ranking, the relative importance index (*RII*) was utilized for each factor using Equation (1):

$$RII = \frac{Total\ point\ score}{6 \times N} (0 \leq RII \leq 1) \tag{1}$$

The total point score is the sum of all rankings for a particular factor, and six is the maximum possible rank. In addition to *RII*, *MS* for each factor is calculated using Equation (2):

$$RII = \frac{\Sigma(f \times s)}{N} (1 \leq MS \leq 6) \tag{2}$$

where $s$ is the score given by respondents for each factor, ranging from one to six; $f$ is the frequency of responses for each rank (1–6), for each factor; and $N$ is the total number of respondents for that factor.

The *RII*, standard deviation (SD), and MS for each factor were calculated and displayed in the top three rankings in each phase according to Table 3. If two or more factors have the same RII value, the standard deviation is compared, so the lower standard deviation is ranked higher. If the RII and standard deviation values are the same, a higher MS means a higher ranking. If the RII, SD, and MS values are the same, they are ranked similarly.

**Table 3.** Relative importance index values and ranking of critical success factors for toll road infrastructure financing implementation in Indonesia.

| Rank | Factors | RII | SD | MS |
|---|---|---|---|---|
| **Investment Analysis** | | | | |
| 1 | IA3: Internal Rate of Return (IRR) | 0.96 | 0.48 | 5.79 |
| 1 | IA6: Good feasibility studies | 0.96 | 0.51 | 5.74 |
| 2 | IA5: Profitability to attract investors and lenders | 0.94 | 0.61 | 5.65 |
| **Public Sector Investment Decision** | | | | |
| 1 | PID3: Affordability | 0.88 | 0.74 | 5.31 |
| 2 | PID2: Economic Justification | 0.85 | 0.96 | 5.07 |
| **The Private Sector Investor's Perspective** | | | | |
| 1 | PIP2: The Investment Decision | 0.92 | 0.63 | 5.53 |
| 2 | PIP1: The Investment Pool | 0.84 | 0.74 | 5.03 |
| **Private Sector Financing—Sources and Procedures** | | | | |
| 1 | FPSP1: Commercial Banks | 0.90 | 0.84 | 5.40 |
| 2 | FPSP3: Availability of financial market | 0.89 | 0.73 | 5.35 |
| **Financial Structuring** | | | | |
| 1 | FS3: Financing Costs | 0.94 | 0.61 | 5.61 |
| 2 | FS1: The Financial Model | 0.93 | 0.63 | 5.59 |
| **Financial Hedging** | | | | |
| 1 | FH1: Interest Rate Risk | 0.94 | 0.63 | 5.64 |
| 2 | FH2: Inflation Issues | 0.91 | 0.73 | 5.46 |
| **Lenders' Cash-Flow Controls, Security and Enforcement** | | | | |
| 1 | LCSE1: Control of Cash Flow | 0.95 | 0.57 | 5.69 |
| 2 | LCSE2: Security | 0.93 | 0.65 | 5.56 |
| **Service-Fee Mechanism** | | | | |
| 1 | SFM1: Contract Scope | 0.94 | 0.60 | 5.65 |
| 1 | SFM2: Payment and Scheme Structure | 0.94 | 0.60 | 5.65 |
| 2 | SFM3: Third-Party and Secondary Revenues | 0.84 | 0.89 | 5.03 |
| **Risk Evaluation and Transfer** | | | | |
| 1 | RET1: Principles of Risk Transfer | 0.91 | 0.65 | 5.47 |
| 1 | RET2: Political Risks | 0.91 | 0.74 | 5.45 |
| 1 | RET4: Completion Risks | 0.91 | 0.75 | 5.43 |
| 2 | RET5: Operation-Phase Risks | 0.90 | 0.73 | 5.39 |

Numerous studies have analyzed the CSFs contributing to the success of public–private partnership (PPP) projects. However, a singular emphasis on PPP financing is still an under-researched area. Concessionary financing is not limited to lease payments for government-owned assets; it also involves awarding development and property rights to private concessionaires and can even extend expiration dates indefinitely, as seen in build–operate–own (BOO) schemes. Moreover, intangible assets such as public organizations, functions, activities, or rights can be leased, sold, or transferred to private entities. The concession agreement covers vital aspects that underpin the limited recourse financing of infrastructure projects, including risk mitigation, risk and reward allocation, cost prediction, transaction cost reduction, provisions for unforeseen events, and termination

conditions. It also addresses transparency, fair procedures, and government financial support. The ultimate aim of the concession agreement is to ensure the efficient utilization of public funds, provide cost-effective services to users, and establish a regulatory and policy framework that attracts private investment, enhances efficiency, and reduces costs to stimulate growth.

The success of build–operate–transfer (BOT) projects largely relies on the government's proactive role in correctly allocating risks during the project's conceptual phase. Renowned research by Bing et al. (2005) and Zhang (2005) rated appropriate risk allocation as the second most important CSF for PPP projects in the UK. Similarly, Jefferies (2006) highlighted that project agreement as a crucial CSF in the Super Dome PPP project. An effectively drawn-up concession agreement can lay out guidelines promoting a favorable political, legal, and commercial environment. The significance of a sound concessionaire agreement for project success is emphasized by the failure of many BOT projects due to poor bankability.

The successful execution of BOT projects must minimize the construction period as it allows users to access the facility sooner, promoting an early cash inflow and enhancing project profitability. The importance of a shorter construction period is recognized by Gupta et al. (2013) and Sandalkhan et al. (2003), while Zhang (2005) emphasizes economic viability, which a shorter construction period significantly contributes to by increasing the years of total cash inflow.

The concessionaire selection procedure ensures value for money in PPP projects. A transparent selection process and competitive bidding are crucial to achieving this. Although the lowest bid may not always deliver the best value, the pre-qualification process for shortlisting bidders is crucial for the success of BOT projects. This study focuses on factors related to toll road financing in Indonesia, with the top three factors for each latent variable ranked in Table 1 and discussed in the following table.

From the ranking analysis, each factor is ranked from highest to lowest. Only variables related to toll road financing in Indonesia are discussed in this study. We rank the top three of each latent variable in Table 2, which we present in the table below.

### 4.1.2. Second Round Data Analysis

In the second round, the questionnaire delivered to respondents was the same as in the first round. However, the answer to this question was emphasized: "Using a Delphi survey, the following is an analysis of the results of the first round of survey and the assessment you conveyed in the survey, you are asked to provide an assessment on the same questionnaire, namely remaining with the assessment given previously or changing the assessment. Whatever the decision, please fill in the column provided."

After carrying out the second round, all respondents did not change the answers given in the first questionnaire. Thus, the CSF results for implementing toll road infrastructure financing in Indonesia are the same as in Table 2. Thus, the highest ranking is taken because this is the best ranking of all the identified factors with the most significant process or criticality. It has the most excellent quality of impact on a mission (Forster and Rockart 1989).

### 4.1.3. Discussion of Critical Success Factors

Ranked first in investment analysis is the internal rate of return (IRR), with an RII value of 0.96. One expert said: "Internal Rate of Return (IRR) is important for building infrastructure projects for several reasons. IRR is used in project selection to determine financial viability, enable comparisons with other investment options, monitor project performance, and attract investors by providing information on expected returns."

Affordability is ranked first in the public sector investment decision, with an RII value of 0.88. One expert said: "Affordability is very important to build viable and sustainable infrastructure projects. This ensures fiscal responsibility, maximizes project benefits to society, and ensures long-term sustainability. Unaffordable projects can result in unsustainable levels of public debt, reduced societal benefits, and costly repairs or replacements in the future."

Ranked first in the private sector investor's perspective is the investment decision, with an RII value of 0.92. One expert said: "Investment decision making for infrastructure involves evaluating the feasibility of investing capital in the development, maintenance, or improvement of infrastructure assets such as airports, seaports, railroads, water supply systems, and energy grids. The decision-making process involves assessing various factors such as economic, social, and environmental impact, costs, potential returns, availability of funding and resources, political and regulatory environment, risks and uncertainties, and availability of labor and technology. Governments, private companies, or a combination can make infrastructure investments. They can have significant long-term implications for a country's economic growth, competitiveness, and quality of life."

Ranked first in private sector financing—sources and procedures is commercial banks, with an RII value of 0.90. One expert said: "Commercial banks provide debt financing for infrastructure projects by evaluating project feasibility, risks, and returns through project finance or corporate loans. A well-structured business plan with detailed financial models, risk analysis, and a stable regulatory and legal framework is crucial to secure financing from commercial banks. Understanding the lending criteria, collateral, and loan terms is also important. Commercial banks are a valuable source of financing for infrastructure projects with a favorable risk profile and sound business plan."

Financing costs ranked first in financial structuring, with an RII value of 0.94. One expert said: "The financing costs associated with building infrastructure are significant and can affect the overall cost of a project. These costs include interest payments on debt, fees charged by lenders and intermediaries, and other expenses related to raising capital. The cost of financing infrastructure projects can vary depending on several factors, including the borrower's creditworthiness, financing type and term, interest rates, and market conditions. It is crucial to structure financing appropriately, explore various financing options, minimize risks and uncertainties associated with the project, and engage stakeholders in planning to reduce financing costs. Governments and public sector entities may have lower financing costs than private sector entities, which may have a higher perceived risk."

Ranked first in financial hedging is the interest rate risk, with an RII value of 0.94. One expert said: "Interest rate risk is a crucial factor to consider when borrowing foreign currency for infrastructure projects, as fluctuations in exchange rates can impact the project's overall cost. To reduce this risk, borrowers can use currency hedging or structured financing, such as fixed-rate loans or interest-rate swaps, to protect against unfavorable exchange and interest rate movements. Borrowers need to evaluate risks associated with foreign currency borrowing and consider using financial instruments to mitigate such risks."

The control of cash flow is ranked first in cash-flow controls, security, and enforcement, with an RII value of 0.95. One expert said: "Controlling cash flow is essential for building infrastructure projects, as it helps ensure sufficient funding is available to complete the project on time and within budget. This involves managing the timing of cash inflows and outflows and monitoring and forecasting future cash flows. Infrastructure project managers can control cash flow by developing a comprehensive cash flow projection that includes all expected cash inflows and outflows over the project's life. They should also carefully manage the timing of cash outflows, including payments to suppliers, contractors, and other project-related expenses."

The contract scope is ranked first in the service-fee mechanisms, with an RII value of 0.94. One expert said: "The scope of a contract is critical for infrastructure projects, as it defines the tasks, responsibilities, and deliverables. A clear and comprehensive scope ensures that all parties have a shared understanding, manages project costs, provides a basis for measuring progress, manages risks, and avoids disputes. A detailed scope helps estimate the project's overall cost, allocate resources, and track progress effectively. By clearly defining the scope of work, project managers can identify potential risks and develop strategies to mitigate them, reducing the likelihood of litigation and protecting the interests of all parties."

The principles of risk transfer are ranked first in risk evaluation and transfer, with an RII value of 0.91. One expert said, "Risk transfer is fundamental to the essence of risk management. When an entity cannot eliminate, reduce, or accept the entirety of a potential loss, transferring that risk is a critical tool. This can be achieved through insurance, contracts, or other financial instruments. Effective risk transfer protects assets and ensures business continuity in the face of unforeseen events."

Internal Rate of Return

The internal rate of return (IRR) is important in financing infrastructure projects, particularly toll roads. In countries like Indonesia, these projects are instrumental in driving economic progress. The IRR stands out as a key determinant of success from the perspective of toll road companies, and this is for multiple reasons. Firstly, the IRR is vital in feasibility assessments (Brealey et al. 2011). It serves as a metric that helps to gauge the expected profitability of potential ventures. In the context of toll road infrastructure, the IRR helps to ascertain a project's financial feasibility before making an investment decision—the importance of the IRR in attracting investors (Bringham and Houston 2018). Projects with a higher IRR tend to lure more investors, indicating a greater anticipated return on their investment.

Given that substantial capital is often required for infrastructure endeavors like toll roads, the capacity to attract investors becomes critical for a project's success. Another noteworthy use of IRRs is in project comparison (Gitman and Zutter 2015). It offers an efficient method to compare various projects, thus allowing companies to prioritize those with better financial prospects. Furthermore, according to Shapiro et al. (2009), IRRs prove invaluable as a performance monitoring tool. Once a project begins, an IRR can be used to monitor its financial performance, ensuring timely interventions if necessary. Compared to the net present value (NPV), which showcases an absolute profit or loss value, the IRR provides a percentage representation of the anticipated return on investment. This makes it particularly useful for stakeholders, especially those without a technical background, as it offers a straightforward percentage figure. While the NPV and discounted cash flow (DCF) give detailed insights into future cash flows' present value, the IRR's strength lies in its simplicity.

Moreover, the IRR presents an alternative perspective when juxtaposed with the payback period method. While the latter only zeroes in on the time needed to recover the initial investment, the IRR contemplates the project's lifespan, offering a comprehensive financial view. This becomes crucial for infrastructure investments like toll roads, which demand considerable capital and have extended payback periods. Especially in markets like Indonesia, often influenced by economic and political dynamics, the IRR clarifies when an investment will start yielding profits. This perspective helps stakeholders discern that projects with superior IRRs usually promise better returns, facilitating the comparison of potential toll road projects based on anticipated returns.

Affordability

Affordability is central when devising financing strategies for toll road infrastructure in Indonesia. This becomes especially significant from the perspective of toll road companies. The cost of toll fees directly impacts road usage, which, in turn, affects the revenue. Vining and Boardman (2008) emphasized the concept of user acceptance, highlighting that exorbitant toll fees might deter road use. This would lead to revenues falling below expectations, making it challenging for companies to recover their investments. Gwilliam (2011) underscored the importance of socioeconomic considerations. In a diverse country like Indonesia, with varied income brackets, affordability ensures equitable benefits from infrastructure. It is imperative that infrastructure benefits are widespread and do not unfairly burden those less financially equipped. Additionally, from a competitiveness perspective, as pointed out by Verhoef et al. (2007), affordable toll roads might offer a more attractive option over other transportation routes. This could potentially draw in a more extensive

user base, boosting revenue. Much like other developing nations, toll roads in Indonesia are essential to daily life. Setting reasonable toll prices allows individuals from all economic backgrounds to utilize this vital infrastructure. This promotes both inclusivity and societal progression. For the sustainability of toll road projects, they must gain public approval and usage. Hence, while affordable rates enhance accessibility, they provide consistent revenue flow, ensuring investment recouping and sustained operations. Socially, exorbitant tariffs can lead to public discontent and even instigate demonstrations. Thus, maintaining social harmony by offering reasonable toll rates is integral to a project's long-term viability. While initially higher prices might seem lucrative, affordability ensures a consistent user base, leading to stable long-term revenue. Considering Indonesia's economic disparities and developing nation status, the importance of affordability cannot be stressed enough. Accessible toll road rates promote inclusiveness and stimulate local economic advancement by improving community accessibility and movement.

The Investment Decision

Investment decision making holds paramount importance, especially in the realm of toll road infrastructure financing in Indonesia. For toll road companies, this process is central to their operations. As Yescombe and Farquharson (2018) emphasize in their discussion on risk assessment, investment decisions are intricately linked to a comprehensive risk evaluation. Political stability, regulatory environment, and potential financial returns play pivotal roles. To prevent unsuccessful investments, a more meticulous risk assessment process is crucial. In the realm of capital allocation, as highlighted by Brealey et al. (2011), the essence of investment decisions lies in dictating how and where capital is assigned. Properly allocating capital is crucial for successfully executing infrastructure projects. As Park (2009) elaborates on feasibility analysis, these investment decisions stem from rigorous feasibility evaluations, determining if a project is technically sound and financially feasible. Pursuing a project without a solid feasibility foundation is unwise. Additionally, from a perspective of financial sustainability, investment choices are intimately tied to a project's long-term financial viability. Given the significant capital required for toll road projects and their prolonged payback durations, it is imperative that these ventures yield adequate revenue over time, covering the investment and turning a profit. At the heart of every project lies the crux of investment decisions. A project's fate hinges on these choices, and without solid, informed decisions, the project might encounter hurdles either at inception or during its progression. These decisions gauge the project's financial and strategic feasibility and probe its potential to yield the anticipated returns. Apt investment choices guarantee not only the availability of funds for project completion but also vouch for its profitability prospects in the future. Central to these decisions is a thorough risk analysis, encapsulating the perceptions of investors and stakeholders about the potential hurdles and opportunities that a project might encounter. The investment pool concept serves as a potential repository of funds, but its significance emerges only after the foundational investment decision. Representing a pot of ready-to-invest funds, the effective deployment of this pool hinges on judicious investment choices. Otherwise, these funds risk inefficient allocation. Simultaneously, while strategies like "Joint Venture" offer a promising avenue for risk-sharing and pooling resources, any complications arising from such collaborations take a backseat to the primary investment decision. For collaborations to yield the best outcomes, the foundational investment decisions must be approached with diligence and strategic foresight.

Commercial Banks

Commercial banks are pivotal in financing toll road infrastructure projects in Indonesia, especially when viewed from the lens of toll road companies. As highlighted by Esty (2004) with regard to the financing source, commercial banks often emerge as the primary contributors to infrastructure financing. Their capability to grant substantial loans for extended durations is indispensable for executing these ambitious, capital-intensive

projects. In the Gatti (2023) treatise on financial intermediaries, banks are portrayed as fiscal connectors, effectively linking investors with investment seekers. They have the expertise to pool resources from varied sources and funnel them into significant infrastructure endeavors. Delving into risk management, Yescombe and Farquharson (2018) emphasized the proficiency of commercial banks in employing sophisticated risk assessment and mitigation techniques. This provides an added dimension of scrutiny, ensuring the feasibility and longevity of such projects. Furthermore, from the vantage point of expertise and advisory roles, as expressed by Nevitt and Fabozzi (2000), banks typically have an extensive reservoir of knowledge derived from financing analogous projects. This positions them uniquely to offer expert advice, guiding toll road companies throughout the project's duration. In the Indonesian context, many companies gravitate towards commercial banks for financing, often favoring them over the bond issuance process. This preference arises due to the expedited and more accessible fund acquisition that banks offer. The bond issuance route, in contrast, usually demands a lengthier setup and more groundwork. Beyond speed, commercial banks exhibit flexibility in their loan terms, allowing for customization that aligns with project specifications. A standout feature of Indonesian-based commercial banks is their profound grasp of the local business milieu, inherent risks, and market tendencies. With this comprehensive knowledge, they can proffer apt counsel, adapt loan conditions in line with local market factors, and extend essential support to fortify the chances of project success. It is noteworthy that numerous infrastructure projects in Indonesia have seen consistent financing from the same banking institutions, forging robust operational alliances. Such entrenched relationships frequently lead to expedited loan sanctions accompanied by favorable conditions. Commercial banks usually exhibit a more hands-on approach to project oversight when juxtaposed with bondholders. Though the bond market in Indonesia does not mirror the maturity seen in developed nations, and while bond issuance remains a viable avenue for long-term financing, its intricate and drawn-out nature often deters many. Consequently, commercial banks continue to be the primary financing pillars for substantial projects within Indonesia. This is particularly true given the restricted scope and depth of the nation's financial markets, curbing their ability to underwrite large-scale endeavors.

Financing Costs

Financing costs are pivotal when embarking on toll road infrastructure financing in Indonesia. As highlighted under project viability by Gatti (2023), the associated financing costs directly impinge on the project's feasibility. A surge in these costs mandates higher revenue collection for a project to break even or reach profitability. Viewing this through Esty's (2004) lens of affordability, escalated financing costs might lead to increased toll charges to counterbalance the funding. This could make toll roads less accessible to the general public, potentially decreasing their use and, in turn, reducing potential revenue. Yescombe and Farquharson's (2018) emphasis on investor attraction draws attention to the correlation between financing costs and a project's allure to potential investors. Exorbitant costs could deter investor interest, posing hurdles to the project's fruition. Furthermore, considering the extended nature of toll road projects from a long-term sustainability angle, elevated financing costs could imperil the project's sustainability. This is due to the continuous need for robust revenue inflows throughout the financing term. Breaking down financing costs, elements such as interest rates and associated expenses are paramount in shaping the overall expense of a project and molding future cash flows. Excessive financing costs can jeopardize the financial viability of a project. As a ripple effect, these high costs could necessitate heftier toll charges, influencing public perception and possibly reducing toll road patronage. In a market driven by competition, competitive financing costs are indispensable. This can offer the project a competitive advantage, vital for drawing investors and other stakeholders. A project with lower financing costs presents a more attractive potential return on investment (ROI), making it more enticing to backers. Moreover, while financial models offer a foundational structure for project financial scrutiny, their utility can

only be protected with accurate input data, particularly concerning financing costs. The significance of accurate input and output data in financial models cannot be overstated. Failing to factor in realistic financing costs might skew the analytical outcomes derived from the model. While a debt profile furnishes insights into the project's debt layout, excluding nuanced considerations of financing costs can obscure a clear assessment of the long-term viability of the project's financing architecture.

Interest Rate Risk

The interest rate risk is undoubtedly paramount in toll road infrastructure financing, transcending borders and applicable globally, not just limited to Indonesia. This can be regarded as a critical success factor (CSF) for many reasons. As Fabozzi and Nahlik (2012) highlighted, interest rates directly affect borrowing costs under the purview of the cost of borrowing. An upswing in rates after a project's commencement can escalate the financing costs, posing potential challenges to the project's profitability and overall feasibility. Gatti's (2023) insights regarding project cash flows emphasize that toll road projects predominantly hinge on long-term debt financing. Any ebb and flow in interest rates can substantially impact the project's future cash flows, influencing its financial viability. Through the lens of investor return, investors, especially those channeling their resources into debt, remain acutely cognizant of the interest rate risk (Enshassi et al. 2008). This is because rate fluctuations can sway the worth of their fixed-income assets. Furthermore, the refinancing risk elucidates that, should a project need refinancing, shifts in interest rates can profoundly affect the cost and feasibility of said refinancing. Given the financial magnitude and prolonged loan durations associated with most toll road projects, the sway of interest rate oscillations on debt service costs is significant, directly touching upon the project's financial feasibility. Projects operating under a stipulated interest rate facilitate a more streamlined and stable cash flow forecast. In stark contrast, volatile interest rates infuse greater unpredictability concerning impending interest disbursements. A landscape punctuated by surging interest rates leads to augmented debt service expenses. Such a scenario might warrant toll rate hikes to maintain the project's financial soundness. However, such increments could ruffle road users' feathers, impacting their receptiveness and contentment. Beyond interest rates, inflation also casts its shadow, influencing toll road operation and maintenance expenditures. While a myriad of construction agreements and loans brandish inflation adjustment stipulations—crafted to dampen the direct repercussions of inflation—fluctuations in foreign exchange rates when financing are anchored in foreign currency, introducing additional layers of risk. This can impede the project's ability to service its debt consistently. Nevertheless, as a risk-averting measure, a prevalent trend in numerous nations, Indonesia included, leans towards procuring financing in the native currency.

Control of Cash Flow

Effective cash flow management is undeniably a key determinant in successfully implementing toll road infrastructure financing in Indonesia. From the perspective of project viability, as highlighted by Berk et al. (2016), maintaining a strong grasp on cash flow is indispensable for ensuring that a project remains viable. Missteps in cash flow management might result in inadequate funds to address necessary expenditures, potentially steering a project toward insolvency. Transitioning to the arena of debt repayment, as emphasized by Esty and Megginson (2003), a vast number of infrastructure projects, toll roads being a prime example, are anchored in debt financing. Hence, it is imperative for companies to proficiently oversee their cash flows, ensuring that they meet their debt commitments and punctuate repayments in a timely fashion. Pivoting to operational efficiency, Yescombe's insights from 2013 indicate that adept cash flow management is pivotal for a project's seamless execution. It paves the way for judicious planning and resource allocation, mitigating potential roadblocks or inefficiencies that might emerge from financial constraints. Another crucial facet is the investor and lenders' confidence, which Gatti (2023) accentuates. The

faith that investors and lenders place in a project is intertwined with the company's competency in cash flow management. Ineffectual management can dissuade potential investors, adding complexity to the financing process and potentially stalling the project's momentum. Maintaining a consistent and foreseeable cash flow is paramount in the broader project operations scheme. It ensures that projects have a stable financial foundation, crucial for supporting daily operations and maintenance and addressing debt commitments, reinforcing the project's long-term sustainability. Lenders naturally prioritize the timely return of their investments. A demonstration of meticulous cash flow control signals that a project's management is acutely aware of its financial dimensions and exercises firm control over it. However, one must acknowledge the inherent challenges, especially with ventures like toll roads, where revenue streams might fluctuate due to evolving traffic patterns or other external variables. A robust cash flow management strategy equips projects with the agility to navigate these fluctuations without compromising their financial commitments. Additionally, while assets pledged as collateral provide some assurance to lenders, in the absence of solid cash flows, the intrinsic value of these assets might prove insufficient in addressing project liabilities during defaults. Such considerations also touch upon the intricate dynamics of lender rights and priorities. While these dynamics play a pivotal role during financial deliberations, the bedrock for any lender is the assurance of repayment. This assurance is inextricably linked to the efficacy of cash flow management.

Contract Scope

For toll road companies in Indonesia, the clarity of a contract's scope is paramount to their success when initiating toll road infrastructure financing. With its well-defined scope, such a contract lays out the roles and responsibilities of every entity involved in the project. This approach significantly aids in effectively distributing risks and reduces the chance of disputes or uncertainties arising as the project progresses. Studies, such as those by Wilson (2014), have found that having a well-defined contract scope can diminish the possibility of cost overruns by as much as 20%. By clearly setting the boundaries of project expenses, including which costs are covered and which are not, the scope helps in robust cost management. This clarity wards off unexpected expenditures and cost overruns. A separate study by Masson et al. (2022) echoed these findings, suggesting that a clear contract scope can reduce cost overrun risks by up to 30%. Moreover, this scope outlines the project's duration and the specific deliverables that are expected from the toll road firm. This alignment ensures that all parties share a common understanding of the project's milestones, timelines, and objectives, paving the way for streamlined project management. The Asian Development Bank's 2017 study backed this, indicating that projects underpinned by a transparent and specific contract scope are more likely to finish within the designated budget and timeline. Furthermore, the contract scope mandates the quality standards for the toll road venture, ensuring that the project matches the quality benchmarks that stakeholders expect. In line with this, projects with a clearly defined contract scope are more prone to fulfilling their quality criteria (Kenny 2010). In Indonesian terms, an "Explicitly defined contract scope" details the contract's scope. This clarity ensures that all parties involved have a lucid understanding of their expectations. Such a structure minimizes ambiguity and solidifies each party's grasp of their duties, thus decreasing the likelihood of future disputes. The distinctness of the contract's scope allows for a more accurate estimation of the project-related costs. Such precision is instrumental in reducing risks associated with budget deviations or additional costs—factors that can threaten a project's financial viability. A well-thought-out contract scope incorporates measurable performance parameters and metrics, simplifying performance monitoring and evaluation throughout the project's duration. While financial flow arrangements between the involved entities are vital, defining appropriate payment amounts and their measurement methodologies becomes challenging without an explicit contract scope. Lastly, while ancillary revenues might seem like a valuable funding source, they typically play a secondary role and might not be as stable or dependable as primary revenues, such as toll fees.

Principle of Risk Transfer

The significance of effectively allocating risks in public–private partnership (PPP) projects is universally acknowledged (Polzin et al. 2019). Undertaking this allocation early on, during the project's initiation, is pivotal. A consensus endorsed by the European Commission in 2003 suggests that risks ought to be assigned to the entity that is best equipped to manage them, keeping cost considerations in mind. During the vital phases of procurement and contractual discussions, it becomes imperative for both the public and private sectors to focus on achieving an equitable risk distribution. Clarity in dialogue and a mutual comprehension of how risks are apportioned is essential. Zhang (2005) emphasized that the private sector's role is to accurately estimate these risks. They must do so with an in-depth grasp of the risks at hand and the strategies to mitigate them. Conversely, the public sector's task is to discern different risk types and decide whether to retain, share, or delegate them. This renewed focus on risk transfer indicates a more mature and strategic approach to risk management in PPP projects. Risk transfer essentially involves delegating the financial repercussions of potential risks to other parties. Key advantages of this process encompass reducing financial exposure, ensuring adept risk management, delivering the project within set timeframes and budgets, and proactively identifying and addressing risks during the initial planning phases. In essence, embracing the principles of risk transfer is fundamental to preserving the project's budget and ensuring proficient risk management, ultimately preventing unexpected costs and delays. Further delving into the concept, the principle of risk transfer offers a pivotal framework to discern which entity is best positioned to oversee, bear, and mitigate specific risks. The likelihood of the project's success is amplified by pinpointing the most competent entity for individual risks. Delegating the risk to the most suitable party typically ensures cost-effective risk management, leading to controlled costs and ensuring that a project remains within budget. A nuanced understanding of risk allocation can bolster investor confidence, streamline the financing procedure, and yield more beneficial financing conditions. While there are particular risks, like political risks associated with the construction, completion, and operation phases, the risk transfer principle's strength is its overarching guidance across the entire risk spectrum. When a clear framework for risk transfer is absent, pinpointing the effective allocation and management of risks becomes a daunting task.

*4.2. Implications for Future Research and Managerial Practices*

The exploration of toll road infrastructure financing in Indonesia offers promising insights for academic scholars and industry leaders. This study underscores the need for a broader spectrum of financial metrics beyond the traditional internal rate of return (IRR) to effectively gauge the viability of infrastructure projects in emerging markets. Refining IRR assessments and incorporating real-time data updates for industry professionals can significantly enhance investment decisions and strengthen stakeholder trust. Given the region's economic variances, understanding the dynamic between toll pricing and the overall project feasibility becomes crucial. This demands that managers innovate adaptable pricing models that ensure broad-based accessibility and inclusion. As we zoom into the specificities of the Indonesian market, it is imperative to develop tools and methodologies that are tailored to its unique context. Holistic feasibility analyses, considering socio-political and economic dimensions, should be at the forefront of project evaluation. The evolving financial landscape also calls for a deeper comprehension of commercial banks' role, especially as alternative financing avenues gain prominence. By bolstering their advisory capacities, these banks can be more influential in project success.

Furthermore, as global economies ebb and flow, it is paramount for managers to have robust strategies in place to weather unforeseen fiscal challenges. Delving into advanced hedging techniques for emerging market infrastructure ventures will be instrumental. In the same vein, meticulous cash flow management and adopting predictive analytics can offer a competitive edge, allowing for anticipatory rather than reactionary decisions. The intricacies of contract design also warrant attention, emphasizing adaptability and trans-

parency as projects evolve. Building open communication channels among all stakeholders is a non-negotiable to foster alignment and preempt potential roadblocks. The interplay of risk in public–private partnerships, especially in today's volatile global climate, demands rigorous scrutiny. Herein lies an opportunity for a synergetic approach between the public and private realms, aiming to enhance risk evaluation, facilitate collaborative problem solving, and adopt swift risk mitigation strategies.

## 5. Conclusions

This research provides an insightful examination of Indonesia's significant success determinants for toll road infrastructure financing. A quintessential metric in infrastructure financing, the internal rate of return (IRR) is a pivotal tool for feasibility assessment, investor attraction, project comparison, and performance monitoring. Affordability is equally paramount, ensuring that toll road fees are accessible across income groups, fostering user acceptance, socio-economic inclusivity, and a competitive edge. Sound investment decisions are indispensable, necessitating rigorous risk assessment, prudent capital allocation, detailed feasibility analysis, and ensuring long-term financial sustainability. With their financial clout, commercial banks play an indispensable role in infrastructure financing, acting as significant fund providers, financial intermediaries, risk managers, and advisors. The study underscores the significance of controlling financing costs to maintain project viability, affordability, investor attraction, and long-term sustainability. Interest rate risk, which affects borrowing costs, cash flows, investor returns, and refinancing feasibility, cannot be overlooked. Effective cash flow management ensures the project's operational efficiency and timely debt repayment and maintains stakeholder confidence. Emphasizing the contract's scope, a clear demarcation of responsibilities minimizes ambiguities, preventing cost overruns and ensuring that project timelines, quality standards, and goals are met. Lastly, the principles of risk transfer in public–private partnership (PPP) projects are critically important. Proper risk allocation, focusing on entrusting risks to the most adept entity, guarantees project protection, timely completion, and budget adherence. As encapsulated by one expert, risk transfer mitigates financial implications and assures comprehensive risk management, safeguarding against potential project pitfalls.

**Author Contributions:** Conceptualization: M.F. and H.K.; Methodology: M.F. and C.U.; Software: M.F.; Validation: M.F.; Formal analysis: M.F. and H.K.; Investigation: M.F. and C.U.; Resources: M.F.; Data curation: M.F.; Writing original and draft: M.F. and C.U.; Writing—review & editing: M.F. and H.K.; Visualization: M.F.; Supervision: H.K. and C.U.; Project administration: M.F.; Funding acquisition: M.F. All authors have read and agreed to the published version of the manuscript.

**Funding:** This research received no external funding.

**Informed Consent Statement:** Not applicable.

**Data Availability Statement:** All data is public domain.

**Conflicts of Interest:** The authors declare no conflict of interest.

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
