# Peer review of "Implementing Toll Road Infrastructure Financing in Indonesia: Critical Success Factors from the Perspective of Toll Road Companies"

_ijfs, doi:10.3390/ijfs11040135_

Round 1

Reviewer 1 Report

Comments and Suggestions for Authors

The literature review is adequate

Objectives. the researchers do not present a clear objective for their research, it is not concrete.

Methodology. The Delphi method is not well formulated, nor is it clear who the interviewees were, what information they provided and how many rounds were conducted to obtain the information.

Author Response

We would like to express my sincere gratitude to you and the referees for the constructive comments and suggestions for our manuscript, ID ijfs-2609141. We have carefully considered each comment and have made the appropriate revisions to address them. Below, we provide a point-by-point summary of the revisions and our responses to the referees' comments:

  1. Referee #1 Comment:

Objectives. The researchers do not present a clear objective for their research, it is not concrete.

Response:

I have written the background of the research .... This research aims to fill the gap regarding identifying critical success factors required for financing toll road infrastructure. Specifically, it aims to craft a comprehensive CSF model from the vantage point of local toll road companies, providing invaluable insights for investors in PPP-based infrastructure projects and enhancing comprehension of Indonesia's unique context......

  1. Referee #2 Comment:

Methodology. The Delphi method is not well formulated, nor is it clear who the interviewees were, what information they provided and how many rounds were conducted to obtain the information.

Response:

I have written it in line number 328 ... Furthermore, the Delphi technique is commonly used for identifying factors in research. It fosters effective group dynamics via an anonymous, multi-step survey process that uses group feedback as a control mechanism after each round (von der Gracht 2012). Delphi surveys have been extensively used in management research for empirical data collection, particularly in situations requiring complex modeling where expert opinion consensus or convergence is vital (Hallowell and Gambatese 2009).. and line number 368 ... Data analysis from the Delphi method carried out in this research was divided into two rounds, namely the first round of data analysis and the second round of data analysis ....

We believe that these revisions have improved the quality and clarity of our manuscript and hope that it now meets the International Journal of Financial Studies standards. We appreciate the time and effort you and the referees dedicated to reviewing our work and look forward to your feedback.

Reviewer 2 Report

Comments and Suggestions for Authors

The paper does a nice job of explaining why Public-Private Partnerships (PPPs) are a good mechanism for developing infrastructure. I'm almost inclined to say it does too nice a job, because it does not discuss the downsides, which may be corruption in the process that determines who gets contracts, and the establishment of state-sanctioned monopolies in the resulting projects. One reason they can be profitable is that they exclude competitors.

An extensive literature on rent-seeking and interest group politics is relevant. Even though that literature rarely discusses PPPs directly, the political process it discusses is directly relevant. The paper's literature review discusses factors related to PPPs, but does not consider the broader political environment, which falls under the heading of public choice.

Some readers might be critical of the survey data the paper uses, but this seems like a good methodology for this project. The paper has surveyed experts in the field, and for projects like this, expert opinion should count for a lot.

The meat of the paper is in section 4, titled "Results and Discussion." The discussion does an excellent job of explaining the results, so much so that it would make a good concluding section of the paper.

Section 4.11 of the paper seems like a better conclusion than the "Conclusion" section of the paper. The concluding section has almost no meaningful content, while Section 4.11 gives policy advice based on the paper's analysis. This is what readers would want to know, and would take from the paper's research. The concluding section could be completely eliminated, because the conclusions readers would care about are nicely discussed in the preceding section.

Author Response

I would like to express my sincere gratitude to you and the referees for the constructive comments and suggestions for our manuscript, ID ijfs-2609141. We have carefully considered each comment and have made the appropriate revisions to address them. Below, I provide a point-by-point summary of the revisions and our responses to the referees' comments:

  1. Referee #1 & #2 Comment:

The paper does a nice job of explaining why Public-Private Partnerships (PPPs) are a good mechanism for developing infrastructure. I'm almost inclined to say it does too nice a job, because it does not discuss the downsides, which may be corruption in the process that determines who gets contracts, and the establishment of state-sanctioned monopolies in the resulting projects. One reason they can be profitable is that they exclude competitors.

An extensive literature on rent-seeking and interest group politics is relevant. Even though that literature rarely discusses PPPs directly, the political process it discusses is directly relevant. The paper's literature review discusses factors related to PPPs, but does not consider the broader political environment, which falls under the heading of public choice.

Response:

I have added a statement regarding the weaknesses of PPP listed on line 73 ... However, PPPs have their downsides. Companies might aim for profit without genuine effort, possibly through favorable government contracts. They might sway officials to frame contracts that do not serve public interests. Numerous stakeholders, from corpo-rations to environmentalists, push for PPP projects, occasionally placing particular in-terests above the public's (Borman and Janssen 2013a). Bureaucrats, motivated by personal interests, significantly influence PPP outcomes. Politicians might favor projects for elec-toral benefits rather than broader societal good. The private sector's deeper insights can lead to imbalanced risk and reward sharing. Moreover, the political landscape, encom-passing governance and transparency, profoundly impacts the success and fairness of PPPs (Gawel 2017).

  1. Referee #3 Comment:

Some readers might be critical of the survey data the paper uses, but this seems like a good methodology for this project. The paper has surveyed experts in the field, and for projects like this, expert opinion should count for a lot.

 Response:

Respondents in the research were people selected based on their expertise and current experience in their field

  1. Referee #4 & #5 Comment:

The meat of the paper is in section 4, titled "Results and Discussion." The discussion does an excellent job of explaining the results, so much so that it would make a good concluding section of the paper.

Section 4.11 of the paper seems like a better conclusion than the "Conclusion" section of the paper. The concluding section has almost no meaningful content, while Section 4.11 gives policy advice based on the paper's analysis. This is what readers would want to know, and would take from the paper's research. The concluding section could be completely eliminated, because the conclusions readers would care about are nicely discussed in the preceding section.

 Response:

The results and discussion have been revised according to reviewer input

We believe that these revisions have improved the quality and clarity of our manuscript and hope that it now meets the standards of the International Journal of Financial Studies. We appreciate the time and effort you and the referees dedicated to reviewing our work and look forward to your feedback.

Thank you for considering our manuscript for publication.

Warm regards,

M Fauzan

Reviewer 3 Report

Comments and Suggestions for Authors

The study explores the CSFs for the implementation of toll infrastructure financing in Indonesia.

My suggestions are as follows:

The authors should review the citations in the paper. A lot of parts of the study need citation. (For example “Across Asia, countless individuals grapple with significant difficulties in accessing vital services. 28 Over 400 million people are without electricity, 300 million lack safe water, and a shocking 29 1.5 billion people are deprived of basic sanitation amenities.” Where do you get these figures? Please check all paper again.)

The authors should write why infrastructure is important for countries through its economic value and comparing the developed and developing countries

The authors should write why they focus on toll infrastructure because there are a lot of infrastructure issues.

Why are Public-Private Partnerships (PPPs) better than other alternatives?

The literature review consists of only a Table. The authors considerably expand this section and uncover the gap they fill in the literature.

You should support with the related literature “The Critical Success Factors (CSFs) approach is a powerful method for identifying 189 key elements vital for effective operation, from managing time and processes to achieving 190 desired results.”

You should benefit from the literature relatively more in discussing your findings.

Conclusions should be remarkably expanded. This section includes your final evaluations, limitation, future research directions. You should summarize your results and give suggestions based on your findings and related literature.

Comments on the Quality of English Language

The language of the study needs a serious improvement.

Author Response

I would like to express my sincere gratitude to you and the referees for the constructive comments and suggestions for our manuscript, ID ijfs-2609141. We have carefully considered each comment and have made the appropriate revisions to address them. Below, I provide a point-by-point summary of the revisions and our responses to the referees' comments:

  1. Referee #1 Comment:

The authors should review the citations in the paper. A lot of parts of the study need citation. (For example “Across Asia, countless individuals grapple with significant difficulties in accessing vital services. 28 Over 400 million people are without electricity, 300 million lack safe water, and a shocking 29 1.5 billion people are deprived of basic sanitation amenities.” Where do you get these figures? Please check all paper again.

Response:

I have added citations to sentences that must be quoted.

  1. Referee #2 Comment:

The authors should write why infrastructure is important for countries through its economic value and comparing the developed and developing countries

Response:

I have added the benefits of infrastructure according to your request.

..In developed countries, advanced infrastructure paves the way for diversified economies, elevated GDPs, and sustainable urbanization. Conversely, many developing nations grappling with inadequate infrastructure often witness constrained growth, reduced foreign investments, and urbanization woes. Infrastructure is much more than bricks and mortar. It is the foundation upon which economies grow, societies thrive, and nations compete globally (Asian Development Bank 2012)...

  1. Referee #3 Comment:

The authors should write why they focus on toll infrastructure because there are a lot of infrastructure issues.

Response:

     I have presented the introduction, by outlining the benefits of PPP and the challenges faced.

  1. Referee #4 Comment:

Why are Public-Private Partnerships (PPPs) better than other alternatives?

Response:

I've written it in the background

... Filling these infrastructure gaps is crucial for improving living conditions and fos-tering sustainable regional growth. The shortage of infrastructure in developing Asia stems from restricted financial resources and effective methods for resource allocation. Despite the acknowledged need for infrastructure advancement, stringent fiscal condi-tions and limited public sector capability impede progress in bridging this gap. One proposed solution involves enlisting the private sector for infrastructure development, utilizing their expertise in operational efficiency, financing, innovation, and skills (Laishram and Kalidindi 2009). Reimagining the collaboration between the private and public sectors through Public-Private Partnerships (PPPs) can improve the adequate provision of public goods and services. PPPs represent long-term contracts where private entities and government bodies collaborate, with the private sector taking on substantial risk and managerial responsibility in exchange for performance-based compensation. The investment in PPPs has the potential to address these ongoing infrastructure challenges...

  1. Referee #5 Comment:

The literature review consists of only a Table. The authors considerably expand this section and uncover the gap they fill in the literature.

Response:

I have described each piece of literature.

  1. Referee #6 Comment:

You should support with the related literature “The Critical Success Factors (CSFs) approach is a powerful method for identifying 189 key elements vital for effective operation, from managing time and processes to achieving 190 desired results.”

Response:

I've added a citation.

  1. Referee #7 Comment:

You should benefit from the literature relatively more in discussing your findings.

Response:

I have collaborated according to your request.

  1. Referee #8 Comment:

Conclusions should be remarkably expanded. This section includes your final evaluations, limitation, future research directions. You should summarize your results and give suggestions based on your findings and related literature.

Response:

I have already presented the conclusions and implications for future research and managerial practices.

We believe that these revisions have improved the quality and clarity of our manuscript and hope that it now meets the standards of the International Journal of Financial Studies. We appreciate the time and effort you and the referees dedicated to reviewing our work and look forward to your feedback. Thank you for considering our manuscript for publication.

Warm regards,

M Fauzan

Round 2

Reviewer 3 Report

Comments and Suggestions for Authors

Dear Authors;

Congratulations!

You sufficiently revised the paper and in turn I suggest "accept in present form" 

Comments on the Quality of English Language

The language of the paper seems satisfactory.